# Push-Pull Effect of Terpyridine Substituted by Triphenylamine Motive—Impact of Viscosity, Polarity and Protonation on Molecular Optical Properties

**DOI:** 10.3390/molecules27207071

**Published:** 2022-10-20

**Authors:** Anna Maria Maroń, Oliviero Cannelli, Etienne Christophe Socie, Piotr Lodowski, Barbara Machura

**Affiliations:** 1Institute of Chemistry, University of Silesia, Szkolna 9, 40-006 Katowice, Poland; 2Laboratory of Ultrafast Spectroscopy (LSU), Lausanne Centre for Ultrafast Science (LACUS), École Polytechnique Fédérale de Lausanne, CH-1015 Lausanne, Switzerland; 3Photochemical Dynamics Group, Institute of Chemical Sciences and Engineering (ISIC), École Polytechnique Fédérale de Lausanne, CH-1015 Lausanne, Switzerland

**Keywords:** 2,2′,6′,2′′-terpyridine, triphenylamine, intramolecular charge transfer, polarity, viscosity, protonation

## Abstract

The introduction of an electron-donating triphenylamine motive into a 2,2′,6′,2′′-terpyridine (terpy) moiety, a cornerstone molecular unit in coordination chemistry, opens new ways for a rational design of photophysical properties of organic and inorganic compounds. A push-pull compound, 4′-(4-(di(4-tert-butylphenyl)amine)phenyl)-2,2′,6′,2′′-terpyridine (tBuTPAterpy), was thoroughly investigated with the use of steady-state and time-resolved spectroscopies and Density Functional Theory (DFT) calculations. Our results demonstrate that solvent parameters have an enormous influence on the optical properties of this molecule, acting as knobs for external control of its photophysics. The Intramolecular Charge Transfer (ICT) process introduces a remarkable solvent polarity effect on the emission spectra without affecting the lowest absorption band, as confirmed by DFT simulations, including solvation effects. The calculations ascribe the lowest absorption transitions to two singlet ICT excited states, S_1_ and S_2_, with S_1_ having several orders of magnitude higher oscillator strength than the “dark” S_2_ state. Temperature and viscosity investigations suggest the existence of two emitting excited states with different structural conformations. The phosphorescence emission band observed at 77 K is assigned to a localized ^3^terpy state. Finally, protonation studies show that tBuTPAterpy undergoes a reversible process, making it a promising probe of the pH level in the context of acidity determination.

## 1. Introduction

**2,2****′:6****′,2****′′-terpyridine** (**terpy**) and its derivatives are among the most important building blocks in coordination and supramolecular chemistry. Three heterocyclic nitrogen donors make **terpy** suitable to form stable coordination compounds by chelating transition metal ions, whereas the rigidity of the **terpy** core facilitates the formation of supramolecular architectures through non-covalent interactions [1]. Since 1932, when **terpy** was first reported by Morgan and Burstall [2,3], a great variety of **terpy** derivatives and their transition metal compounds have been synthesized and characterized in numerous applications [1,4,5,6,7,8,9,10,11,12,13,14,15,16,17,18,19,20]. Several studies revealed that the variation of substituents attached to the **terpy** unit causes an impressive change in the photophysical and electrochemical properties of these ligands and of their complexes [13,21,22,23,24,25,26]. Most importantly, the introduction of strong electron donating groups into the electron acceptor **terpy** core gives rise to Intraligand Charge Transfer (ICT) transitions, which offer a large degree of control over the system’s optical properties [27,28]. Other remarkable aspects of these compounds are their optical sensitivity to the pH, polarity and viscosity of the local environment [29,30,31,32,33,34,35,36,37,38,39,40,41].

Among possible electron donating units, particular attention was devoted to **triphenylamine** (**TPA**), which is well known for its light-harvesting capabilities, high hole mobility properties, and non-coplanar structure preventing molecular aggregation [42,43]. Organic push-pull systems based on the **TPA** motive are of high importance for potential applications in solar-energy conversion [44,45,46,47,48,49,50] and as organic light-emitting diodes [51,52,53,54], logic gates [55], field-effect transistors [56,57,58,59] and photoswitches for theranostics [60,61,62]. In our previous contribution, we demonstrated the beneficial impact of the **TPA** unit on the luminescent properties of **terpy** and its Re(I) coordination compound [63]. **4′-(4-(di(4-tert-butylphenyl)amine)phenyl)-2,2′,6′,2′′-terpyridine** ligand (**tBuTPAterpy**) (Figure 1) was found to be highly emissive in chloroform solution (λ_PL_ = 489 nm, Q.Y. = 84%), whereas the Re(I) carbonyl complex [ReCl(CO)_3_(tBuTPAterpy)], in which the **tBuTPAterpy** is coordinated to the metal centre in a bidentate way, showed significantly enhanced photoluminescence (PL) in comparison with [ReCl(CO)_3_(terpy-κ^2^N)] (see also Appendix A in Electronic Supporting Information (ESI)).

In the present work, ICT processes in **tBuTPAterpy** are thoroughly investigated with steady-state and time-resolved optical spectroscopy and Density Functional Theory (DFT) simulations. The in-depth knowledge of the system’s photophysical properties is of high significance for the rational design of new functional materials with tailored optical behavior.

## 2. Results and Discussion

### 2.1. Optical Spectra of tBuTPAterpy vs Constituent Building Blocks

The absorption and emission properties of the push-pull **tBuTPAterpy** molecule were compared to those of its constituent building blocks considering the combination of two possible pairs: (i) 2,2′:6′,2′′-terpyridine (**terpy**) and bis(4-tert-butylphenyl)aniline (**tBuTPA**), and (ii) 4′-phenyl-2,2′:6′,2′′-terpyridine (**4****′-Ph-terpy**) and bis(4-tert-butylphenyl)amine (**tBuDPA**) (see Appendix A). To minimize the impact of the solvent polarity, the absorption and emission spectra of **tBuTPAterpy** and its building blocks were measured in the apolar solvent *n*-hexane, and the results are shown in Figure 1.

The UV-Vis spectrum of **tBuTPAterpy** in Figure 1a shows two well-resolved bands, with maxima at 291 nm and 364 nm, and with the high energy absorption profile fairly reproduced by the sum of the absorption bands of **terpy** and **tBuTPA**. The low-energy absorption band of **tBuTPAterpy** at 364 nm is absent in all model chromophores, meaning that it arises from the conjugation of the two building blocks **terpy** and **tBuTPA** into the extended molecule (**tBuTPAterpy**), most likely involving a charge transfer process from the electron-rich **tBuTPA** donor to the electron-deficient **terpy** acceptor. The fluorescence spectrum of **tBuTPAterpy**, in black in Figure 1b, cannot be ascribed to the emission of any of the building blocks. In fact, this emission band is distinctly red-shifted with respect to the others, suggesting that it originates from an electronic state that is more delocalized than the separated molecular moieties.

### 2.2. Solvent Polarity Effect

Steady-state electronic absorption and emission spectra of **tBuTPAterpy** were recorded in a wide range of solvents of different polarities in order to characterize their photophysical properties as a push-pull system. The spectra are shown in Figure 2, Figure 3 and Figure 4 and summarized in Table 1 (see also Appendix A in the ESI) [64,65,66].

The position of the **tBuTPAterpy** absorption maxima in Figure 2 is marginally affected by the solvent polarity, which, however, modulates the molar absorption coefficients of the entire spectrum and the width of the low-energy band (see also Appendix A in the ESI). Specifically, by going from non-polar to polar solvents, a decrease of the extinction coefficients and an increase of the Full Width at Half Maximum (FWHM) is observed, indicating a low polar character of the Ground State (GS) [68,69,70,71].

In contrast, the fluorescence spectra of **tBuTPAterpy** reported in Figure 3 show a strong dependence of the position of the emission band on the solvent polarity [72]. The emission maximum progressively shifts to longer wavelengths with increase the solvent polarity, namely from 407 nm in *n*-hexane to 527 nm in acetonitrile. This significant red-shift is accompanied by changes in the emission profile—from narrow and vibronically structured in apolar *n*-hexane and cyclohexane to structureless and very broad (FWHM of 4050 cm^−1^) in the more polar aprotic acetonitrile. Such solvatochromic behavior is typical of push-pull systems undergoing a photoinduced ICT process, which leads to the formation of a highly polar emitting state that is stabilized by polar solvents with respect to the neutral GS [70,71].

To estimate the difference between the excited and ground state dipole moments (Δ*µ* = *µ*_e_ − *µ*_g_), the correlation between the solvent polarity and the Stokes shift was analysed using Lippert equations [73]:

(1)$$h\nu_{Em}=h\nu_{o}^{Em}-\;\frac{2\mu_{e}\left(\mu_{e}-\mu_{g}\right)}{a_{o}^{3}}\times f\left(\varepsilon ,n\right)$$(2)$$h\nu_{Abs}=h\nu_{o}^{Abs}-\frac{2\mu_{e}\left(\mu_{e}-\mu_{g}\right)}{a_{o}^{3}}\times f\left(\varepsilon ,n\right)$$
where: $h\nu_{Em}$ represents the emission energy of the compound in a particular solvent; $h\nu_{o}^{Em}$ corresponds to the absorption and emission energies in vacuum, $\mu_{g}$ and $\mu_{e}$ are the dipole moments of the molecule in its ground and excited states, $a_{o}$ is the Onsager cavity radius, and f is defined as:(3)$$f\left(\varepsilon ,n\right)=\frac{\varepsilon -1}{2\varepsilon +1}-\frac{1}{2}\left(\frac{n^{2}-1}{2n^{2}+1}\right)$$

The plot of the Stokes shift of **tBuTPAterpy** against the orientation polarizability (Figure 3, top inset) shows a very good linearity, spanning from 2887 cm^−1^ in *n*-hexane to 8609 cm^−1^ in acetonitrile, as expected for a push-pull compound. The large Stokes shift values in polar environments can be ascribed to a remarkable change in the dipole moment of **tBuTPAterpy** between the GS and the emitting excited state [74], with an estimated Δ*µ* value of 23.49 D, assuming the Onsager cavity radius of 6.79 Å determined by quantum chemical calculations.

The solvent also affects the excited state lifetime of **tBuTPAterpy** at room temperature (Table 1 and Appendix A), which gradually increases with the solvent polarity. Concerning the quantum yield of the system, it is very high in chloroform, ethyl acetate, tetrahydrofuran, dichloromethane, dimethylformamide, dimethylsulfoxide (0.7–0.84), and it is slightly attenuated in acetonitrile (0.63) and in the apolar solvents *n*-hexane (0.48), cyclohexane (0.54), toluene (0.64). Instead, a dramatic reduction occurs in methanol (0.02), which is probably related to conformational changes induced by H-bonding interactions between the terpyridine nitrogen atoms and the solvent molecules.

The impact of the solvent polarity on the emission spectra of **tBuTPAterpy** was also investigated in rigid matrices formed at the liquid nitrogen temperature. Under these conditions, the solvent reorganisation effect is strongly reduced, limiting the spectral tuning of the ICT states [75]. As shown in Figure 4, the emission energy of **tBuTPAterpy** at 77 K is indeed slightly affected by the solvent polarity. In this condition, the estimated value of Δ*µ* is reduced to 14.02 D.

Compared to the room temperature measurements, the fluorescence maximum of **tBuTPAterpy** in *n*-hexane and cyclohexane red-shifts upon cooling to 77 K. An opposite trend occurs for solvents with an *E*_T_(30) larger than 33 kcal·mol^−1^. For these media, the low-temperature emission maximum is systematically narrowed and blue-shifted compared to the room-temperature spectra. This finding is compatible with a stronger ICT character of the emitting state upon solvent polarity increase because the rigidity of the 77 K glassy matrix prevents the full stabilization of the solute through solvent reorganization.

It is worth highlighting that the temperature decrease leads to higher emission intensities while the corresponding excited state lifetimes become shorter. For instance, the PL decay time of **tBuTPAterpy** in butyronitrile is twice longer at room temperature than that at low temperature (Appendix A). This observation agrees with the hypothesis that the initially populated Franck–Condon (FC) state and the Lowest Energy Excited State (LEES) have different electronic characters.

### 2.3. Temperature and Solvent Viscosity Effects

To investigate the impact of the solute conformational changes on its photophysics, steady-state PL spectra were acquired in a methanol:ethanol mixture (1:4) at selected temperatures in the 80–290 K range (Figure 5a). Additionally, the **tBuTPAterpy** room temperature emission was measured in several mixtures of two solvents having similar polarity, but different viscosity, glycerol (η = 954 cP; ε = 46.5)/methanol (η = 0.54 cP; ε = 32.7), and Time-Resolved Emission Spectra (TRES) were recorded in pure glycerol, the solvent with the highest viscosity, with the results respectively reported in Figure 5b,c.

Figure 5a shows that the emission band of **tBuTPAterpy** remains centred at ~430 nm and slightly changes in intensity in the temperature range from 80 to 110 K, i.e., when the intramolecular rotations of the solute are hindered and the solute-solvent interactions are weak. Upon temperature increase up to 210 K, a bathochromic shift of the band is observed along with a gradual drop of its intensity, suggesting that conformational changes in the electronically excited state of **tBuTPAterpy** become allowed, stabilizing the ICT emitting state. For temperatures higher than 230 K, the emission energy and its intensity remain unchanged.

Stationary **tBuTPAterpy** PL spectra collected in methanol: glycerol mixtures of variable composition are reported in Figure 5b and show that the addition of glycerol up to 30% induces a small decrease of the emission intensity at 560 nm. Instead, in the 20:80 methanol: glycerol mixture, the emission band centered at 560 nm shifts towards higher energies, and a strong band appears at 450 nm. A further increase of the glycerol proportion up to 90% leads both emission bands to slightly red-shift and a significantly increase in intensity [72]. Finally, in pure glycerol (η = 954 cP at 25 °C), **tBuTPAterpy** is characterized by two emission bands centered at 458 nm and 594 nm, with intensities respectively higher and lower than in the 10:90 methanol: glycerol mixture, and shows an isoemissive point at 517 nm. The appearance of a second emission band upon solvent viscosity increase indicates the presence of two emitting states having different structural conformations.

TRES of **tBuTPAterpy** recorded in pure glycerol as a function of the time delay (Figure 5c) also report the presence of an isoemissive point around 20 ns, suggesting that the two emissive states are populated in a sequential way upon a conformational change. In less viscous solvents, this structural modification probably occurs on shorter time scales, and it cannot be observed due to the limited time resolution of the TRES measurements (see Appendix A), calling for dedicated ultrafast investigations in order to characterize this population transfer process.

### 2.4. Protonation Effect

The effect of protonation on the absorption and emission properties of **tBuTPAterpy** was studied in titration experiments conducted with the use of **trifluoroacetic acid** (**TFA**) in CHCl_3_ using the procedure previously described in [34]. The portions of TFA were selected in order to cover a broad range of titration steps from 1:1 to 1:1000. Then, the deprotonation experiment was conducted with the use of the strong basis **triethylamine** (**TEA**), adding 1000 equivalents to the 1:1000 **tBuTPAterpy:TFA** sample.

Upon the addition of **TFA** to the chloroform solution, naked eye color changes from pale yellow to orange-red are observed. The changes in the absorption and emission profiles of **tBuTPAterpy** are shown in Figure 6a–c and Appendix A of the ESI. In the absorption spectra (Figure 6a), the gradual addition of **TFA** (1–1000 equivalents) leads to an intensity decrease of the band at 372 nm and to the formation of two new bands with maxima at 330 nm and 493 nm. In the literature, the band at 330 nm was ascribed to ^1^π → ^1^π* transitions of the protonated **terpy** unit [76,77]. Instead, the spectral red shift in the visible region was attributed to the enhancement of the **tBuTPAterpy** ICT character due to the electron-withdrawing increase of the **terpy** acceptor upon protonation [34]. Isosbestic points at 352 nm and 398–434 nm indicate the presence of multiple protonated–neutral forms in equilibrium between each other.

The PL spectra in **tBuTPAterpy** titrated with **TFA** acid (Figure 6b and Appendix A) show significant quenching of the fluorescence band at 476 nm upon the increase of the acid concentration. Starting from the addition of 30 equivalents of **TFA** (inset in Figure 6b), the decrease in the emission band at 476 nm occurs together with the appearance of a red-shifted emission band. The spectra for mixtures **tBuTPAterpy**:**TFA** with acid fractions higher than 200 equivalents exhibit only one emission maximum (at 595 nm for a 1:200 mixture and at 624 nm for a 1:1000 one). The PL lifetime for the protonated form is 5.92 ns, and it is 1.4 times higher than in the neutral compound (see Appendix A in the ESI). The presence of an equilibrium between neutral and protonated forms of **tBuTPAterpy** is supported by the observation of an isoemissive point around 520–530 nm, which is visible in the normalized steady-state emission spectra of the system upon the addition of **TFA** (1–1000 equivalents) (Appendix A in the ESI), as well as in the TRES collected for the 1:50 **tBuTPAterpy**:**TFA** mixture (Figure 6c). The excitation spectrum recorded for the protonated form displays a maximum at 435 nm (see Appendix A in the ESI), which well overlaps with the isosbestic point of the absorption spectra.

Finally, the reversibility of the protonation/deprotonation processes was demonstrated by recovering the original absorption and emission spectra of neutral **tBuTPAterpy** after the addition of 1000-equivalents of **TEA** to the final mixture of **tBuTPAterpy** and **TFA** (1:1000) (Figure 6a,d and Appendix A) [78].

### 2.5. Phosphorescence of tBuTPAterpy

The phosphorescence of **tBuTPAterpy** was measured and compared to the triplet emission of the model chromophores **terpy** and **tBuTPA**, as reported in Figure 7a,b. The 77 K steady-state emission spectrum of **tBuTPAterpy** was recorded with the addition of a 10% dopant of ethyl iodide—a fluorescence quencher—in order to promote phosphorescence. Having a heavy iodine atom, ethyl iodide facilitates the intersystem crossing via a stronger spin-orbit coupling [79]. The results in Figure 7a show that the low-temperature phosphorescence band of **tBuTPAterpy** overlaps with its room-temperature ICT emission. In Figure 7b, the phosphorescence spectra of **tBuTPAterpy**, **terpy** and **tBuTPA** are respectively observed in the ranges: 460–650 nm, 425–650 nm and 400–550 nm. By comparing the phosphorescence of **tBuTPAterpy** with its model building blocks, we conclude that the triplet state of **tBuTPAterpy** is predominately related to the electron-acceptor **terpy** fragment. Also, low-temperature TRES of **tBuTPAterpy** in BuCN highlights the late appearance (>27 ns) of an emission band in the same wavelength range of the isolated **terpy** unit, suggesting the formation of a localised **terpy** triplet state (Figure 7c).

### 2.6. Quantum Mechanical Calculations

Figure 8 shows the simulated absorption spectra of **tBuTPAterpy** in the wavelength range 240–500 nm that were obtained from the calculated vertical excitation energies and the relative oscillator strengths between singlet states. The position of the simulated band maxima agrees well with the experiment (differences within 10–25 nm) and has little dependence on the solvent polarity. An even better agreement between experiment and theory is found for the position of the band at ~390 nm (differences of only a few nm).

In all solvents, the absorption band for wavelengths longer than 350 nm involves two electronic transitions with significantly different oscillator strengths. Specifically, the excitation to the lowest singlet state, S_1,_ has an oscillator strength several orders of magnitude higher than the excitation to the S_2_ state. As such, the absorption band is dominated by an electronic transition to the lowest singlet state, S_1_, while the state S_2_ acts as a “dark” state, not being involved in the light absorption process. The second band at shorter wavelengths (about 300 nm for the simulated spectra) arises from the transition to four further singlet states, three of which are characterized by a significant oscillator strength, while the transition to the S_3_ state is one order of magnitude less intense than the others (Appendix A). Except for S_6_, all these states result from an electronic excitation from the Highest Occupied Molecular Orbital (HOMO), which is localized on the central phenyl ring and on the aromatic rings of the amine substituents, towards the unoccupied π* antibonding orbitals located on different parts of the molecule (Appendix A).

The Lowest Unoccupied Molecular Orbital (LUMO) and LUMO+1 are populated upon transition to the S_1_ and S_2_ electronic states, respectively. Even though the character of both excitations is essentially the same and can be classified as CT, the LUMO orbital extends over both the central phenyl ring and the terpyridine fragment, while the LUMO+1 orbital is located solely on the terpyridine motif (Figure 9). The higher oscillator strength of the S_0_ → S_1_ excitation with respect to the S_0_ → S_2_ one should thus be related to the larger overlap between the two conjugated π-molecular orbitals involved in the former transition. In contrast, the absence of such an overlap in the case of the LUMO+1 orbital results in an almost zero value of the transition moment, leading to an inactive S_0_ → S_2_ excitation. Appendix A shows that the calculated oscillator strength of the HOMO → LUMO transition is drastically reduced upon the increase of the dihedral angle between the plane of the phenyl ring and the plane of the substituent. This effect can be rationalized in terms of a decoupling of the π-electronic system between the central phenyl ring and the substituents, which changes the form and the local symmetry of the HOMO and LUMO orbitals. Specifically, when the plane of the terpyridine rings is perpendicular to the plane of the phenyl ring, the π-orbitals of the phenyl-terpyridine units become completely decoupled from each other, and the oscillator strength drops to almost zero. Furthermore, the LUMO orbital localizes on the terpyridine motif and increases in energy, becoming the LUMO+1 orbital, as shown in Figure 10. As a consequence, the S_1_ and S_2_ states change in their relative energy order, and the oscillator strength of the S_0_ transition towards both states becomes very small.

Since the S_1_ excitation is characterized by a significant value of the oscillator strength, the position of the first absorption band maximum is mostly determined by this electronic transition. In the calculations, the negligible solvent dependence of the S_1_ transition energy agrees very well with the experimental observations, suggesting that both the optimized GS and the FC region of the S_1_ state have comparable solvation energies [72]. This is confirmed by the Polarizable Continuum Model (PCM) energy stabilization of the GS (FC S_1_) state compared to the energy of the isolated system, corresponding to 2.6 kcal/mol (7.1 kcal/mol), 5.3 kcal/mol (7.1 kcal/mol) and 7.6 kcal/mol (13.9 kcal/mol) for *n*-hexane, chloroform and acetonitrile, respectively. Since for all solvents both GS and FC S_1_ states are stabilized by a similar degree, the energy gap between them does not significantly depend on the solvent polarity.

In order to characterize the properties of the emissive states, the electronic and geometric configurations of the system were computed, performing structural optimizations of its relaxed lowest singlet (S_1_) and triplet (T_1_) states by using TD-DFT and Unrestricted Density Functional Theory (UDFT), respectively. The results for the relaxed S_1_ state are shown in Figure 11, while the theoretical values of emission wavelength, oscillator strength and orbital character are reported in Appendix A for both S_1_ and T_1_ states. The de-excitation from the S_1_ state preserves the same CT character (Figure 11b) of the vertical excitation from the relaxed S_0_ structure (HOMO π_R1/Ph_ → LUMO π*_R2/Ph_), and it is independent of the solvent employed in the PCM model. The structural changes of the system as a result of the S_1_ optimized geometry compared to the initial GS structure are mostly related to the spatial arrangement of the amine and terpyridine fragments with respect to the plane of the central phenyl ring (Figure 11a). In the case of the terpyridine group, the dihedral angle between the plane of the substituent and the plane of the phenyl ring becomes close to zero, i.e., both fragments of the molecule lay in a common plane. At the same time, the amino group undergoes a larger torsion with respect to the plane formed by the two other fragments. Therefore, in the relaxed S_1_ geometry, the LUMO orbital is localized on the flattened structure. Instead, the HOMO orbital is almost completely localized along the vertical plane of the amine group with respect to the central phenyl ring. With respect to the ground state, the S_1_ energy can be stabilized, among other processes, by lowering the energy of the LUMO orbital, i.e., by fully coupling the π-electron system between the terpyridine motif and the central phenyl ring. This is achieved when both fragments of the molecule lie in a common plane. At the same time, upon excitation, the depopulation of the HOMO facilitates the rotation of the amine group. It is worth noting that the oscillator strength for the vertical transition from the relaxed S_1_ state is halved compared to the vertical excitation of the absorption process occurring from the relaxed S_0_ state. According to the present analysis, in the relaxed S_1_ state, the decoupling of the HOMO π-electron system that results from the rotation of the amino group (Figure 11b) is the most likely cause of the electronic dipole moment reduction for the excited-to-ground state transition.

The calculated vertical de-excitation energies S_1_ → S_0_, without taking into account the state-specific equilibrium solvation of the excited state, show a negligible dependence on the solvent polarity and correlate quite well with the experimental data for low-temperature PL measurements. When including the contribution of the state-specific equilibrium solvation in the PCM model, the calculated vertical de-excitation energies S_1_ → S_0_ reproduce the relative bathochromic shift observed in the room temperature PL measurements (Appendix A, results indicated by the superscript a), even though the predicted emission energies are systematically underestimated. This inaccuracy can be related to the limitations of the PBE0 functional. However, this functional is expected to correctly describe the electronic structure and the energies of low-lying excited states, as in the case of similar molecular systems [80], since it captures the relative energy shift of the emission experiment for the *n*-hexane, chloroform, and acetonitrile series. Indeed, even though in **tBuTPAterpy** the differences between the calculated emission wavelengths and the experimental results range from ~77 nm to ~109 nm, the relative shift of the simulations correlates well with the experiment. As such, the observed bathochromic shift can be interpreted in terms of the “classic” non-equilibrium state effect of the solvent occurring in the electronic transition between the relaxed excited state S_1_ and the ground state S_0_. Overall, the difference in the equilibrium and non-equilibrium solvent effects originates from the electronic structure of the S_1_ state with respect to the S_0_ state. Compared to the ground state, the geometry relaxation of the S_1_ state leads to a discernible change in the form of the orbitals involved in electronic excitation. The HOMO → LUMO excitation, in combination with the geometry relaxation, increases the electron density polarization, for which one electron remains on the HOMO orbital localized on the amine group, and the excited electron occupies the LUMO orbital located on the remaining part of the molecule. Since the excited electron density increases the difference of the solvent equilibrium interaction around the molecule in the relaxed S_1_ and GS, it also causes a stronger stabilization of the S_1_ state upon solvent polarity increase.

The theoretical analysis presented above allows us to qualitatively explain the experimental results showing the existence of two emitting conformers, one having higher oscillator strength and a blue-shifted emission, the other having lower transition probability and a red-shifted emission. As shown in Figure 5a,b, the **tBuTPAterpy** emission band decreases in intensity and red-shifts upon reduction of the conformational hindrance of the solution by temperature increase and viscosity decrease, respectively. These observations can be rationalized by accounting for the conformational-dependent energy stabilization of the S_1_ state, which involves the rotation of the amine and terpyridine units with respect to the central phenyl ring. As discussed above, this structural modification red-shifts the emission wavelength and decreases the oscillator strength compared to the emission process occurring from the FC S_1_ state, which takes place from a molecular structure corresponding to the S_0_ energy-stabilized structure.

## 3. Materials and Methods

**tBuTPAterpy** was obtained according to the procedure previously described in [63]. The analytical data (^1^H and ^13^C NMR spectroscopy, FT-IR technique, HR-MS and elemental analysis) for **tBuTPAterpy** are reported in ESI and are in good agreement with those reported in ref. [63]. All solvents were of spectroscopic grade, commercially available and used without further purification.

Steady-state electronic absorption measurements were carried out with an Evolution 220 (ThermoScientific, Waltham, MA, USA) spectrophotometer. An FLS-980 fluorescence spectrophotometer (Edinburgh Instruments, Livingston, UK) was used to collect: steady-state emission spectra, TRES and PL lifetime measurements, the latter made using a time-correlated single photon counting (TCSPC). The experimental details of the PL measurements are briefly described in the ESI.

All calculations were obtained employing the DFT [81] and TD-DFT [82], using the hybrid PBE0 functional [83,84] and the def2-SVP basis set [85]. Solvation effects on the system’s geometry and electronic structure were included by using a PCM [86]. Three solvents of increasing polarities were considered in the PCM model: *n*-hexane, chloroform and acetonitrile. At the DFT level, the geometry of the ground state (S_0_) was fully optimized without any structural parameter constraints. The optimized geometry of the ground electronic state was employed in TD-DFT calculations to compute the energies of twenty singlet vertical excitations. The geometry of the lowest singlet excited state (S_1_) was fully optimized using TD-DFT. Full geometry optimization of the lowest triplet state (T_1_) was also performed using the UDFT formalism. The vertical de-excitation energy of the triplet state was determined as the energy difference with respect to the energy of the ground state in the optimised T_1_ geometry.

## 4. Conclusions

By the conjugation of **terpy** and **tBuTPA**, new absorption and emission features appear in **tBuTPAterpy**. The intramolecular charge transfer character of these new bands is determined based on a solvent polarity investigation. At room temperature, the large red-shift and broadening of the emission band upon solvent polarity increase highlights the charge transfer nature of the lowest in energy excited state. At the same time, the negligible dependence of the absorption maximum suggests different characters between the initially populated Franck–Condon state and the lowest energy-emitting state. This conclusion agrees well with the emission measurements at 77 K, which show only minor changes as a function of the solvent polarity. The observation of two emission bands in the PL spectra collected for a series of mixtures of methanol: glycerol of variable composition suggests the presence of two different emitting states. This hypothesis is further supported by the detection of an isoemissive point in the TRES spectra in pure glycerol. Rigidochromic and viscosity effects indicate that the two emitting states are related to conformational changes of the solute. Finally, **tBuTPAterpy** shows reversible acidochromic properties in chloroform solution. DFT calculations reveal a dominant role of S_0_ → S_1_ vertical transition in the absorption band at the lowest energy, which has a charge transfer character. A second S_0_ → S_2_ transition is predicted at similar energies but is not observed in the absorption spectrum because of its much smaller oscillator strength compared to the S_0_ → S_1_ transition. The striking difference in their transition moment values is a consequence of the different overlap between the orbitals involved in the electronic excitation. Moreover, by calculating the potential energy curves for the three lowest singlet states as a function of θ and ϕ dihedral angles, an exchange of the order of the S_1_ and S_2_ states is observed. By comparing the results for vertical transitions starting from the energy minimum of the S_0_ and S_1_ states, the same HOMO π_R1/Ph_ → LUMO π*_R2/Ph_ character is found. Calculations including state-specific equilibrium solvation effects in the PCM provide a theoretical explanation for the substantial solvatochromic spectral shift of the emission band. The two emissive states observed in the experiments correspond to the energy-stabilized S_1_ and Franck–Condon S_1_ states, differing in the relative orientation of the amine-phenyl-terpyridine units. The results reported in this manuscript indicate that **tBuTPAterpy** possesses an intriguing and complex photophysics that is highly sensitive to external conditions. The strong solvatochromism of the presented molecule could be used in applications for bioimaging and biosensing, such as microenviromental polarity and viscosity sensors. Furthermore, the reversible acidochromism of this molecule makes it a potential candidate for volatile acids sensors. Further studies should be devoted to the characterization of the ultrafast dynamics of this molecule upon intramolecular charge transfer excitation.

## Data Availability

Not applicable.

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
