# Peer review of "Push-Pull Effect of Terpyridine Substituted by Triphenylamine Motive—Impact of Viscosity, Polarity and Protonation on Molecular Optical Properties"

_molecules, 2022, doi:10.3390/molecules27207071_

Round 1

Reviewer 1 Report

In the present work, Authors submitted an ICT processes in tBuTPAterpy with steady-state and time-resolved optical spectroscopy and Density Functional Theory (DFT) simulations. An in-depth knowledge of the photoinduced excited state properties in such systems is of high significance for a rational design of new functional materials with a tailored photophysical behavior.

1- NMR spectra of compounds should be given in Suppl. such as previos literature. 

2- Some new references can be added on DFT, ICT and fluorescent compounds. Abnormal push-pull benzo[4,5]imidazo[1,2-a][1,2,3]triazolo[4,5-e]pyrimidine fluorophores in planarized intramolecular charge transfer (PLICT) state: Synthesis, photophysical studies and theoretical calculations, Dyes and Pigments 204, 2022, 110405. Synthesis and photophysical properties of modifiable single, dual, and triple-boron dipyrromethene (Bodipy) complexes, Tetrahedron Letters 56 (14), 1873-1877. Viscosity sensitive semisquaraines based on 1, 1, 2-trimethyl-1H-benzo[e]indole: Photophysical properties, intramolecular charge transfer, solvatochromism, electrochemical and DFT studyJournal of Molecular Liquids, 285, 1 2019, 123-135. A Bodipy-bearing pillar [5] arene for mimicking photosynthesis: Multi-fluorophoric light harvesting system, Tetrahedron Letters 59 (20), 1958-1962. A fluorescent clever macrocycle: Deca-bodipy bearing a pillar [5] arene and its selective binding of asparagine in half-aqueous medium.

3- The vibrations of Fig.5c, 6c and 7c should be decrease with higher slits. 

4- FT-IR specra should be given in Supl. such as previos literature. 

5- Some parameters such as concentration, excitation wavelength, etc. should be given in the explanation of figures. 

6- Why did use TFA and TEA as 1000 equivalents? please explain in. text. 

Best Regards.

Reviewer 2 Report

Authors continues to study the push-pull compound  tBuTPAterpy they previously described as a ligand in Re(I) carbonyl complex. A wide range of properties of this compound has been investigated. The authors used the most modern methods for studying the photophysical properties of the compounds.

Nevertheless, in my opinion, it is worth adding to the conclusions the possible applications of tBuTPAterpy and the features that may arise in this case in the light of the features of tBuTPAterpy discovered by the authors.

The rest of the comments are technical in nature:

- line 13: a space in "A push-pull" is missed;

- lines 30, 53: compound names at the beginning of sentences should be capitalized, even if they are preceded by numbers;

- on the figures 5c, 6b, 6c, 7a-c (as well as on almost all figures in ESI) the captions are too small;

- line 34: the publication cited is from 1932, not 1931;

- line 76: one of maxima is 271 nm, but in Table 1 and ESI the same maximum is 291 nm;

- line 108: the maximum  in acetonitrile is 528 nm, but in Table 1 the same maximum is 527 nm;

-perhaps it is worth adding a table to the ESI with the values indicated in the figures 1a,b.

Also, to facilitate the perception of the material, it is worth inserting the connection structure into the main text of the manuscript.

Round 2

Reviewer 1 Report

accept